# Roles of lung-recruited monocytes and pulmonary Vascular Endothelial Growth Factor (VEGF) in resolving Ventilator-Induced Lung Injury (VILI)

**Chin-Kuo Lin**[1,2], **Tzu-Hsiung Huang**[3], **Cheng-Ta Yang**[4,5], **Chung-Sheng Shi**[2,6]*

**1** Division of Pulmonary Infection and Critical Care, Department of Pulmonary and Critical Care Medicine, Chiayi Chang Gung Memorial Hospital, Puzi City, Taiwan, **2** Graduate Institute of Clinical Medicine Sciences, College of Medicine, Chang Gung University, Taoyuan, Taiwan, **3** Department of Respiratory Therapy, Chiayi Chang Gung Memorial Hospital, Puzi City, Taiwan, **4** Department of Thoracic Medicine, Taoyuan Chang Gung Memorial Hospital, Taoyuan, Taiwan, **5** Department of Respiratory Therapy, College of Medicine, Chang Gung University, Taoyuan, Taiwan, **6** Division of Colon and Rectal Surgery, Department of Surgery, Chiayi Chang Gung Memorial Hospital, Puzi City, Taiwan

* csshi@mail.cgu.edu.tw

**Data Availability Statement:** All relevant data are within the paper and its Supporting Information files.

## Abstract

Monocytes and vascular endothelial growth factor (VEGF) have profound effects on tissue injury and repair. In ventilator-induced lung injury (VILI), monocytes, the majority of which are Ly6C$^{+high}$, and VEGF are known to initiate lung injury. However, their roles in post-VILI lung repair remain unclear. In this study, we used a two-hit mouse model of VILI to identify the phenotypes of monocytes recruited to the lungs during the resolution of VILI and investigated the contributions of monocytes and VEGF to lung repair. We found that the lung-recruited monocytes were predominantly Ly6C$^{+low}$ from day 1 after the insult. Meanwhile, contrary to inflammatory cytokines, pulmonary VEGF decreased upon VILI but subsequently increased significantly on days 7 and 14 after the injury. There was a strong positive correlation between VEGF expression and proliferation of alveolar epithelial cells in lung sections. The expression pattern of VEGF mRNA in lung-recruited monocytes was similar to that of pulmonary VEGF proteins, and the depletion of monocytes significantly suppressed the increase of pulmonary VEGF proteins on days 7 and 14 after VILI. In conclusion, during recovery from VILI, the temporal expression patterns of pulmonary growth factors are different from those of inflammatory cytokines, and the restoration of pulmonary VEGF by monocytes, which are mostly Ly6C$^{+low}$, is associated with pulmonary epithelial proliferation. Lung-recruited monocytes and pulmonary VEGF may play crucial roles in post-VILI lung repair.

**Funding:** This work was supported by the Chang Gung Medical Foundation of Taiwan (grant no. CMRPG6C0231 and CMRPG6C0232) and CKL received the grants. The URL of the foundation website is https://www.cgmh.org.tw/en/. The funders had no role in the study design, data collection and analysis, decision to publish, or preparation of the manuscript.

**Competing interests:** The authors have declared that no competing interests exist.

# Introduction

Mechanical ventilation is essential for the life support of patients with respiratory failure. Nevertheless, it has potential drawbacks that often induce lung damage and lead to ventilator-induced lung injury (VILI). Earlier studies have explored the pathophysiological mechanisms of VILI occurrence and identified inflammatory mediators and cytokines that participate in the initiating insult [1–3]. However, few studies have focused on the temporal expression patterns of inflammatory cytokine and growth factor expression during resolving from VILI and elucidated their associations with subsequent lung repair. Many growth factors are involved in the process of epithelial repair of injured lungs [4]. These are chiefly members of the epithelial growth factor and fibroblast growth factor families. However, it is uncertain whether vascular endothelial growth factor (VEGF) contributes to tissue repair. As one of the crucial angiogenic growth factors, VEGF promotes not only the proliferation of vascular endothelial cells but also maintains survival and regulates the proliferation of alveolar epithelial cells in injured lungs [1, 5–7]. In hyperoxia-induced acute lung injury (ALI), treatment with VEGF enhances healing of lung structure during late recovery [8]. Nevertheless, with respect to VILI, the contribution of pulmonary VEGF to the repair of injured lungs has not been investigated.

Circulating monocytes are derived from the mononuclear phagocytic cells population in the bone marrow. Monocytes, the heterogeneous cells that perform various physiological activities, can be divided into two subsets [9]. In mice, monocytes are classified according to the differential expression of surface markers, including CCR2, CD62L, and $CX_3CR1$ [10]. Mouse monocytes that express CCR2 are known as inflammatory monocytes, while those not expressing CCR2 are termed resident monocytes. Gr1, as well as Ly6C, which is part of the Gr1 epitope, are additional markers of $CCR2^+$ monocytes [10, 11]. The different physiological activities of the monocytes in the vasculature are closely linked to the Gr1 expression (Ly6C) [12–14]. Monocytes play dual roles in tissue injury and repair of the heart and liver [15, 16]. $Ly6C^{+high}$ monocytes participate in injury progression, whereas $Ly6C^{+low}$ monocytes help tissue healing [15, 16]. In VILI, high-stretch mechanical ventilation promotes pulmonary margination of activated $Gr1^{high}$ monocytes and contributes to the lung injury progression [17]. Our earlier study showed that the monocytes recruited to the injured lungs are primarily $Ly6C^{+high}$ and can produce VEGF to regulate vascular permeability during the first couple of hours following VILI [18]. However, the phenotypes of lung-recruited monocytes during recovery from VILI have not been clarified.

In this study, we explore the post-VILI expression patterns of inflammatory cytokines and growth factors, and identify the phenotypes of monocytes recruited to the lungs during recovery from VILI to elucidate how monocytes recruited to lungs and pulmonary VEGF benefit epithelial proliferation.

# Materials and methods

## Animals and two-hit model of VILI

All experimental protocols using animals were performed following the Guide to the Care and Use of Experimental Animals and approved by the Institutional Animal Care and Use Committee of Chang Gung Memorial Hospital, Chiayi, Taiwan (Approval number: 2012102301 (September 21, 2013)). All animal experiments were performed in the Animal Center of Chang Gung Memorial Hospital at Chiayi, and mice were housed following IACUC, national, and international animal welfare protocols. The center introduced the Association for Assessment and Accreditation of Laboratory Animal Care International (AAALAC) certification system in 2013 and had obtained AAALAC certification in July 2014 (No: TTQ08384). To

optimize environmental conditions and reduce stress, animals were housed in a room maintained with low noises and vibrations and under constant temperature (20–25˚C), humidity (40–60%) and a 12-h light/dark cycle; cage occupancy was limited to five mice per cage, where animals were provided with enrichment toys and given free access to food and water. Male C57BL/6 mice were obtained from the National Laboratory Animal Center (Taipei, Taiwan). A two-hit VILI model was used, according to a previously established protocol [18, 19]. The model consisted of the first hit of a subclinical dose of lipopolysaccharides (LPS) to prime circulating inflammatory cells and the second hit of a high tidal volume (HTV) mechanical ventilation to generate VILI. Briefly, mice were anesthetized using intraperitoneal Zoletil 50 (80 mg/kg; Tiletamine-Zolazepam, Virbac, Carros CEDEX, France) and received a single intravenous tail vein injection of 20 ng LPS (O111B4; Sigma-Aldrich, St Louis, MO, USA) immediately before mechanical ventilation [17]. The mice were intubated by inserting a sterile 20-gauge intravenous cannula through the vocal cords into the trachea and ventilated with a SAR-830 series small animal ventilator with multi-animal setups (CWE Inc., Ardmore, PA, USA) in room air and under general anesthesia with regular intraperitoneal injections of Zoletil 50 (10 mg/kg/h) for 4 h. The ventilator was set at a HTV of 40 mL/kg to produce high-stretch mechanical ventilation with an inspiratory time of 0.3 s, a respiratory rate of 80 breaths/min, and end-expiratory pressure (PEEP) of 0 cm $H_2O$. After ventilation, the day 0 mice were sacrificed immediately after administering an overdose of anesthetic (Zoletil 50, 50 mg/kg), while the others were extubated to allow resumption of spontaneous breathing and recovery. The extubated mice received post-treatment care following the IACUC guidelines. We checked the mice every day and measured their body weight twice per week. A full-time veterinarian took care of animal health and well-being until the mice were sacrificed with an overdose of anesthetic on days 1, 3, 7, or 14 after VILI. Mouse lungs were dissected and prepared for further experiments. The numbers of animals used in the experiments and the numbers of animals that died during mechanical ventilation or recovery are detailed in S1 Fig. The number of animals used per group was as follows: normal control (n = 6), day 0 (n = 6), day 1 (n = 6), day 3 (n = 6), day 7 (n = 12, six animals were used for studying the depletion of pulmonary monocytes), and day 14 (n = 12, six animals were used for studying the depletion of pulmonary monocytes).

### Histopathology and immunohistochemistry analysis

For histopathology, mouse lungs were manually inflated and fixed by injecting 10% buffered formalin and embedded in paraffin. Tissues were then sliced (4 μm) and stained with hematoxylin and eosin (H&E). The number of inflammatory cells in the alveoli was determined by counting 100 randomly sampled alveoli under a light microscope at a magnification of 400×. For immunohistochemistry (IHC) analysis, formalin-fixed, paraffin-embedded tissue sections were incubated overnight at 4˚C with anti-VEGF antibodies (VEGF A-20: sc-152, 1:40; Santa Cruz Biotechnology, USA) or anti-Ki67 antibodies (ab15580, 1:100; Abcam, USA) using DAKO antibody dilution buffer (DAKO). The tissue sections were then treated with tetramethylrhodamine isothiocyanate-conjugated (Jackson ImmunoResearch Laboratories) or horseradish peroxidase (HRP)-conjugated secondary antibodies (DAKO) in the same buffer for 1 h at room temperature. In the tissue sections treated with HRP-conjugated antibodies, 3, 3' diaminobenzidine (DAB) (DAKO) was added as the chromogen; hematoxylin or 4',6-diamidino-2-phenylindole (DAPI) (DAKO) was used to counterstain the nuclei. To measure the correlation between pulmonary VEGF expression and epithelial cell proliferation, the number of Ki67-positive cells was counted under a light microscope (Olympus BX51) at a magnification of 400× in 20 consecutive alveoli, and the staining intensities of positive VEGF secretions

were analyzed using Pax-It quantitative image analysis software (Paxcam). Images were standardized for light intensity. For H&E analysis, the number of animals used per group was as follows: normal control (n = 4), day 0 (n = 3), day 1 (n = 3), day 3 (n = 3), day 7 (n = 5), and day 14 (n = 3). For IHC analysis, the number of animals used was n = 6 for each group.

### Preparation of lung single cells and flow cytometric analysis

Lungs were dissected, rinsed in a petri dish containing PBS (pH 7.2) on ice, and cut into pieces of approximately 1–2 $mm^2$. The tissues were milled by mechanical disruption and incubated in tubes with a digestion solution containing Dispase II and DNase I (Sigma) in a water bath at 37˚C for 45 min. The digested lung tissues were filtered through a 100-μm cell strainer and then forced through a 40-μm filter. The cell suspension was treated with 10× RBC lysis buffer (BioLegend) to remove RBCs and suspended in 1 mL staining buffer. Before flow cytometry, the cells in suspension were counted using a hemocytometer. Cells ($10^5$–$10^6$) in 100 μL were dispensed into each tube. Cells were stained with rat anti-mouse CD11b FITC (eBio 11-0112), rat anti-mouse F4/80 PE (eBio 12-4801), and rat anti-mouse Ly6C APC (eBio 17-5932-82) antibodies. The specimens were incubated at 4˚C for 40 min in the dark. The stained cells were analyzed using a FACSCanto flow cytometer (Becton Dickinson, San Jose, CA) and the acquired data were analyzed using FlowJo software (Treestar, Ashland, OR, USA). A FACS-Canto instrument and FACSDiva software (Becton Dickinson, San Jose, CA) were used for flow cytometric analysis, and the acquired data were analyzed with FlowJo software (Treestar, Ashland, OR, USA). For flow cytometric analysis, the number of animals used per group was as follows: normal control (n = 4), day 0 (n = 4), day 1 (n = 4), day 3 (n = 4), day 7 (n = 5), and day 14 (n = 5).

### Quantitation of pulmonary growth factors and cytokines

The concentrations of VEGF, transforming growth factor-beta (TGF-β), interleukin 6 (IL-6), interleukin 1-beta (IL-1β), and tumor necrosis factor-alpha (TNF-α) expressed by lung cells were measured by performing ELISA using the following commercial kits: mouse VEGF Quantikine ELISA Kit, cat# MMV00; mouse/rat/porcine/canine TGF-β Quantikine ELISA Kit, cat# MB100B; mouse IL-1β/IL-1F2 DuoSet ELISA, cat# DY401-05; and mouse TNF-α DuoSet ELISA, cat# DY410-05 (R&D Systems, Minneapolis, MN, USA) and mouse IL-6 ELISA Ready-SET-Go, cat#: 88–7064 (eBioscience, San Diego, CA, USA). ELISA plates were read on a multimode plate reader (BioTek, Winooski, VT, USA). For ELISA, the number of animals used per group was as follows: normal control (n = 4), day 0 (n = 4), day 1 (n = 4), day 3 (n = 4), day 7 (n = 6), and day 14 (n = 6).

### Isolation of lung-recruited monocytes

For PCR experiments, lung-recruited monocytes were obtained by negative selection and isolated using an EasySep Mouse Monocyte Isolation Kit (Stemcell Tech) and EasySep magnet following the manufacturer's instructions. Briefly, lung cells were suspended and prepared at a density of $1 \times 10^8$ cells/mL in PBS/5% FBS. Cells were incubated in a mixture of 50 μL/mL normal rat serum and 50 μL/mL EasySep Mouse Monocyte Enrichment Cocktail at 4˚C for 15 min. Cells were then washed with RoboSep Buffer and centrifuged at 300 ×g for 10 min at 4˚C. The cell pellet was resuspended at the same density and incubated under the same conditions as described above. Subsequently, cells were mixed with EasySep D Magnetic Particles at 150 μL/mL cells and incubated at 4˚C for 10 min. The cell suspension volume was adjusted to 2.5 mL with EasySep Buffer without rat serum. Finally, the centrifuge tube was placed into an EasySep Magnet for 5 min.

Subsequently, the supernatant, in which cells were enriched, was transferred to a new tube, centrifuged at $300 \times g$ for 10 minutes, and resuspended at $1\times10^8$ cells/mL in the recommended medium. The cell suspension was mixed and incubated with species-specific FcR blocking antibody for mouse cells at 10 μL/mL and anti-macrophage antibody [BM8] at a final concentration of 0.3–3.0 μg/mL at room temperature for 15 min. EasySep PE Selection Cocktail was added to a concentration of 100 μL/mL cells and incubated at room temperature for 15 min. EasySep magnetic nanoparticles were added to a concentration of 50 μL/mL cells and incubated at room temperature for 10 min. The cell suspension was adjusted to 2.5 mL by adding the recommended medium. The tube was placed into the magnet and set aside for 5 min. The desired fraction was collected into a new 5 mL polystyrene tube contained monocytes.

## RT-PCR analysis of VEGF mRNA

Total RNA was extracted from the isolated monocytes using TRIzol reagent (Invitrogen, Life Technologies) following the manufacturer's protocol. Five nanograms of total RNA was reverse-transcribed using the SuperScript III First-Strand Synthesis System (Invitrogen, Life Technologies) and following the manufacturer's protocol. Real-time PCR was performed to determine VEGF mRNA expression levels, using glyceraldehyde 3-phosphate dehydrogenase (GAPDH) as internal control, on a BioRad iQ iCycler Detection System (BioRad Laboratories, Ltd) using SYBR green fluorophore. The primer sequences used in this experiment were as follows:

VEGF forward: `5'-ATCATGCGGATCAAACCTCACCA-3'`
VEGF reverse: `5'-TCACCGCCTTGGCTTGTCACAT-3'`
GAPDH forward `5'-CTCACTGGCATGGCCTTCCGTGT-3'`
GAPDH reverse: `5'-GATGTCATCATATTTGGCAGGT-3'`.

Reactions were performed in a total volume of 25 μL, including 12.5 μL 2× QuantiFast SYBR Green PCR Master Mix (Applied Biosystems), 0.5 μL of each primer (10 mM), and 0.5 μL of reverse-transcribed cDNA. PCR was performed for 10 min at 95˚C to activate the enzyme, denaturation for 10 s at 95˚C, annealing for 15 s at 60˚C, and extension for 10 s at 72˚C for 50 cycles. Relative quantification was conducted using the ΔΔCt method. Results were normalized to the amplified housekeeping gene GAPDH. For RT-PCR analysis, the number of animals used per group was as follows: normal control (n = 4, day 0 (n = 4), day 1 (n = 4), day 3 (n = 4), day 7 (n = 5), and day 14 (n = 5).

## Depletion of pulmonary monocytes

Monocytes were depleted using liposomal clodronates, Clophosome-A (FormuMax Scientific). Some of the mice that were sacrificed on day 7 were injected on day 3 with a single 200 μl dose of Clophosome-A in the tail vein after administering anesthesia with intraperitoneal Zoletil 50 (80 mg/kg); some of the mice that were sacrificed on day 14 received the same dose of Clophosome-A on day 3 and were reinjected with 100 μL dose after the anesthesia on days 7 and 11. Mouse lungs were isolated for VEGF ELISA, as described above.

## Statistical analysis

Statistical evaluations were performed using PASW Statistics version 22. Data were expressed as means ± SD. The normality of the data was examined by Shapiro-Wilk test. For comparisons between two groups, data were analyzed by unpaired Mann–Whitney U test. For comparisons between multiple groups, one-way ANOVA with Bonferroni's or Dunnett's post hoc test or Kruskal-Wallis test with corrections by Bonferroni for multiple testing was used as appropriate. Spearmen's correlation was used to evaluate the relationship between the IHC intensity

of VEGF and the number of Ki67-positive cells in the alveolar epithelium of lung sections. Statistical significance was defined as $p < 0.05$. Detailed illustrations of the statistical analysis are provided in the S1 Text.

## Results

### Abundance of inflammatory cells in lung tissues following VILI

C57BL/6 mice that had received HTV ventilation after LPS treatment were used to determine how VILI influenced the recruitment and abundance of inflammatory cells in lung tissues. H&E staining revealed that immediately after VILI (day 0), the number of inflammatory cells in the injured lung tissues increased and was significantly higher than that in the lung tissues of the normal control group ($45 \pm 3$ vs. $6 \pm 1$ cells per 100 alveoli, respectively; $p < 0.001$, Fig 1). The number of inflammatory cells peaked on day 1 after VILI ($53 \pm 1$ cells per 100 alveoli; $p < 0.001$ compared to the normal control group, and $p < 0.01$, compared to day 0 and day 3 groups, Fig 1). The number of inflammatory cells gradually decreased from day 3 to day 14, but remained higher than that of the normal control group ($39 \pm 4$ to $14 \pm 2$ cells per 100 alveoli, $p < 0.01$, Fig 1).

### Phenotype of monocytes recruited to the lungs during the recovery phase following VILI

Flow cytometry was conducted to identify leukocytes recruited to the lungs and determine the phenotypes of monocytes during recovery after VILI. Lung-recruited leukocytes, which express CD11b, were selected from the lung single-cell suspension (Fig 2A). Monocytes in this leukocyte population were further selected based on the expression of F4/80. Finally, the monocytes were gated and divided into two populations based on the expression intensity of Ly6C. We found that the number of CD11b$^+$ leukocytes recruited to the injured lungs increased following VILI and reached a peak level that was significantly higher than that of the normal control group, on day 1 after VILI ($4.5 \times 10^6 \pm 6.3 \times 10^5$ vs. $2.8 \times 10^4 \pm 1.7 \times 10^4$ leukocytes per lung; adjusted $p = 0.005$, Fig 2B and 2C). However, the number of CD11b$^+$ leukocytes declined and returned to the baseline after day 3 ($9.7 \times 10^4 \pm 4 \times 10^4$ leukocytes per lung; adjusted $p = 1$, compared to the normal control group, Fig 2B and 2C). During recovery, from day 1 to day 14 after VILI, most monocytes recruited to the lungs were Ly6C$^{+low}$ (Fig 2B and 2D).

### Time courses of pulmonary growth factor and inflammatory cytokine expressions during recovery from VILI

The expression of growth factors and inflammatory cytokines in lung tissue lysates following VILI were determined by ELISA. Following VILI, the concentration of interleukin 6 (IL-6) increased from the baseline of $586.2 \pm 189.6$ pg/g to $15817.5 \pm 703.5$ pg/g ($p = 0.006$; adjusted $p = 0.125$, Fig 3A). However, the level decreased by 55% to $7170.8 \pm 151.3$ pg/g on day 3 after VILI (Fig 3A) and returned to the baseline on day 7 and 14 ($1414.9 \pm 900.9$ and $399.5 \pm 158$ pg/g, respectively; adjusted $p = 1$ for both comparisons against the normal control group, Fig 3A). Interleukin-1 beta (IL-1β) and tumor necrosis factor-alpha (TNF-α) also showed similar expression patterns as IL-6. IL-1β concentration increased from $1485.4 \pm 988$ to $8861.8 \pm 517.9$ pg/g (adjusted $p = 0.008$, Fig 3B) and TNF-α concentration increased from $21293.8 \pm 2792.6$ to $52728.1 \pm 4400.4$ pg/g ($p < 0.001$, Fig 3C) on day 1 after VILI but gradually decreased from day 1 on and reached baseline levels on days 7 and 14 (Fig 3B and 3C).

**A**

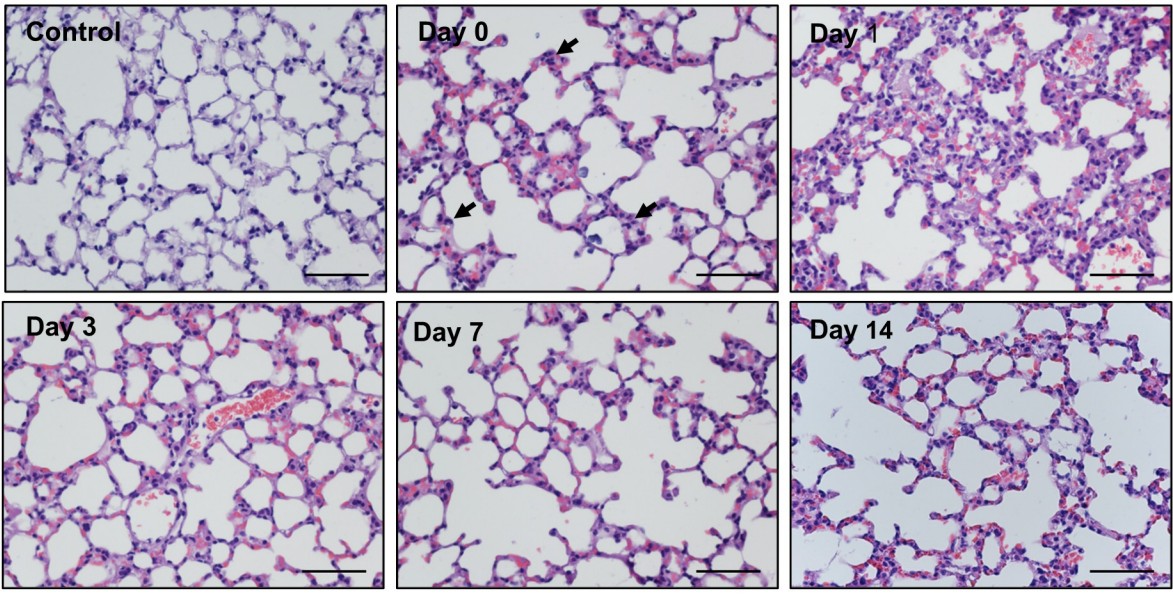

**B**

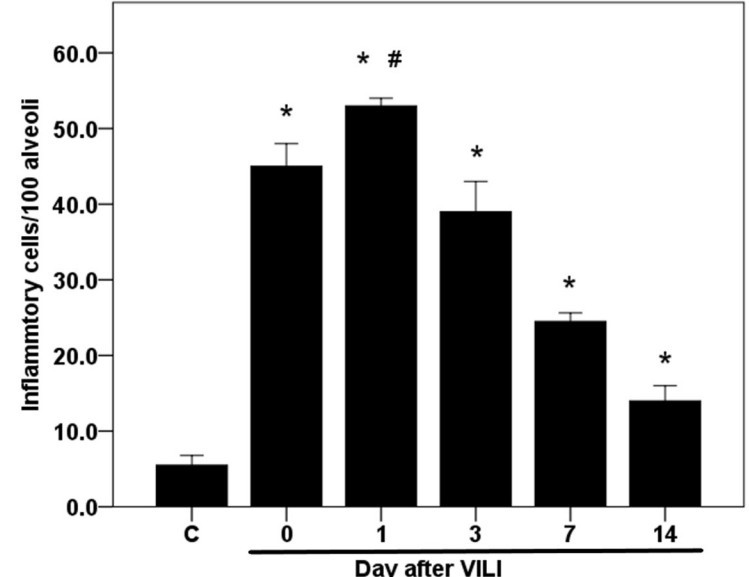

**Fig 1. Time course of lung inflammatory cell response following VILI as determined from histological analysis.** C57BL/6 mice received 20 ng LPS intraventricularly and HTV ventilation to produce a two-hit VILI. The mice were sacrificed immediately after VILI (day 0) or on days 1, 3, 7, or 14 after VILI. (A) H&E staining of the lung sections of mice sacrificed at different time points (original magnification: 200×). The closed arrow indicates lung-recruited inflammatory cells, including neutrophils and monocytes. (B) Quantification and comparison of the inflammatory cells in lung sections from individual mice sacrificed at different time points. Scale bar = 200 μm. Statistical analysis was performed using one-way ANOVA with Bonferroni's post hoc test. Data are expressed as means ± SD; n = 3–5 per group, * $p < 0.01$ vs. normal control (C) and # $p < 0.01$ vs. day 0 and day 3.

Unlike the cytokines, the growth factors tested, tumor growth factor-beta (TGF-β) and VEGF, exhibited a different temporal pattern of expression. TGF-β concentration did not change significantly immediately after VILI at day 0 and remained unchanged until day 1

**A**

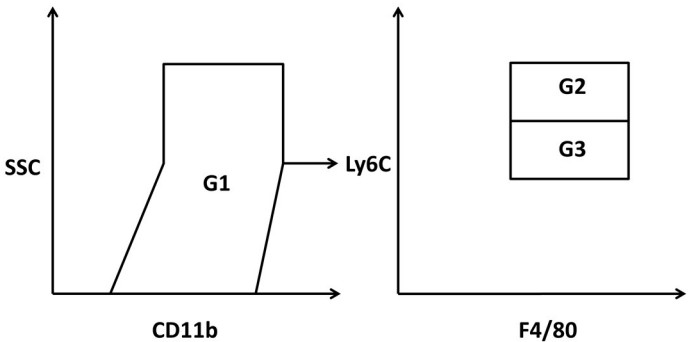

**B**

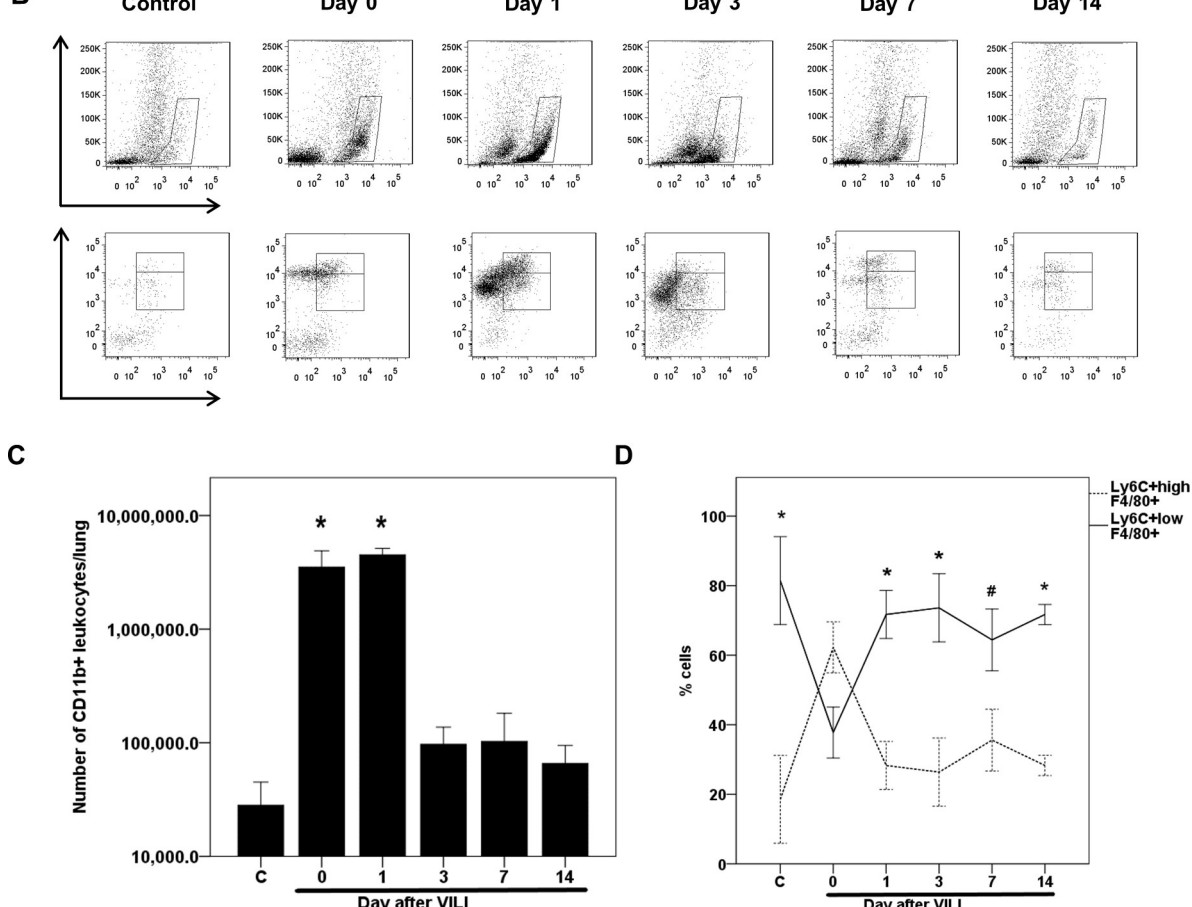

**Fig 2. Phenotype and time course of post-VILI levels of lung-recruited monocytes as determined from flow cytometric analysis.** (A) Gating strategy for differentiating CD11b[+] leukocytes and Ly6C[+] monocyte subsets. CD11b[+] population (G1) was gated for estimation of lung-recruited leukocytes (neutrophils and monocytes). In the G1 population, the F4/80 positive population, which represented monocytes, was gated further and divided into the Ly6C[+high] (G2) and the Ly6C[+low] (G3) population. (B) The time course of lung-recruited leukocytes and the Ly6C phenotypic change of lung-recruited monocytes immediately after VILI (day 0) or on days 1, 3, 7, and 14 after VILI. (C)

Quantification and comparison of lung-recruited CD11b$^+$ leukocytes of individual mice sacrificed at different time points. Statistical analysis was performed using the Kruskal-Wallis test with Bonferroni correction for multiple tests. Data are expressed as means ± SD; n = 4–5 per group, $^*p < 0.05$, vs. normal control (C). (D) The percentages of pulmonary Ly6C$^{+high}$ and Ly6C$^{+low}$ monocytes of individual mice sacrificed at different time points. Statistical analysis was performed using one-way ANOVA with Bonferroni's post hoc test. Data are expressed as means ± SD; n = 4–5 per group, $^*p < 0.001$ and $^\#p = 0.002$, vs. day 0.

post-VILI (Fig 3D). VGEF concentration showed a non-significant decrease from the baseline (8190.2 ± 505.6 to 5276.6 ± 95.2 pg/g, $p = 0.169$ and adjusted $p = 1$, Fig 3E) on day 1 after VILI. On day 3, the concentrations of both growth factors increased compared to day 1 and continued to increase progressively until day 14. On day 14, TGF-β concentration increased to 11542.3 ± 117.2 pg/g (adjusted $p = 0.009$, compared to the normal control group, Fig 3D) and VEGF concentration reached 13755 ± 138.5 pg/g ($p = 0.005$; adjusted $p = 0.099$, Fig 3E) compared to the normal control group. Both concentrations were significantly higher than those at day 0 and day 1 (Fig 3D and 3E).

## Correlation between pulmonary VEGF and alveolar epithelial repair

To assess the role of pulmonary VEGF in post-VILI alveolar epithelial repair, we performed IHC staining to estimate the expression of VEGF and the number of Ki67-positive cells in the alveolar epithelium of lung sections and evaluated the correlation between each measure (Fig 4A and 4B). Consistent with the ELISA results, the relative staining intensity of VEGF decreased by 69% from the baseline (252.8 ± 25.9 to 77.3 ± 7.3) on day 1 after VILI, although this difference did not reach statistical significance ($p = 0.005$ and adjusted $p = 0.075$, Fig 4C). The levels rebounded on day 3, and the relative intensity of VEGF staining increased to a level significantly higher on days 7 and 14 than on day 1 (991.8 ± 119.9 and 2165.8 ± 245.5; adjusted $p = 0.001$ and $<0.001$, respectively, Fig 4C). Similar to the expression pattern of VEGF, the number of Ki67-positive cells decreased by 44% from the baseline (87 ± 5 cells per 100 alveoli to 48 ± 5 cells per 100 alveoli) on day 1 after VILI, however, this decrease was not statistically significant ($p = 0.018$ and adjusted $p = 0.276$, Fig 4D); levels of Ki67-positive cells were higher on day 3 (84 ± 5 cells per 100 alveoli) and reached significantly higher levels on days 7 and 14 than observed on day 1 (106 ± 4 and 216 ± 36 cells per 100 alveoli; adjusted $p = 0.003$ and $< 0.001$, respectively, Fig 4D). There was a strong correlation between the IHC relative staining intensities for VEGF and the number of Ki67-positive cells in the alveolar epithelium (Spearmen's correlation coefficient rho = 0.93, $p < 0.001$, Fig 4E).

## Contribution of monocytes to pulmonary VEGF production during recovery from VILI

In the normal lung, pulmonary VEGF is mainly derived from alveolar epithelial cells [20]. However, large amounts of alveolar epithelial cells are destroyed during VILI, which reduces their contribution to pulmonary VEGF restoration during the recovery phase. To elucidate the role of lung-recruited monocytes in post-VILI restoration of pulmonary VEGF, we analyzed VEGF mRNA expression in lung-recruited monocytes following VILI, calculated the relative change in mRNA, and compared to the untreated control. We found that VEGF mRNA expression decreased immediately following VILI (0.19 ± 0.05 fold as compared to normal control at day 0; $p = 0.009$, Fig 5) and further decreased by 96% relative to the normal control on day 1 after VILI (0.04 ± 0.02 fold; $p = 0.008$, Fig 5). However, levels gradually returned to the baseline after day 7 (0.68 ± 0.22 fold; $p = 0.32$, Fig 5). The temporal expression pattern was consistent with that of pulmonary VEGF protein, as determined by ELISA. To ascertain whether monocytes are a cellular source of pulmonary VEGF during recovery from VILI, we

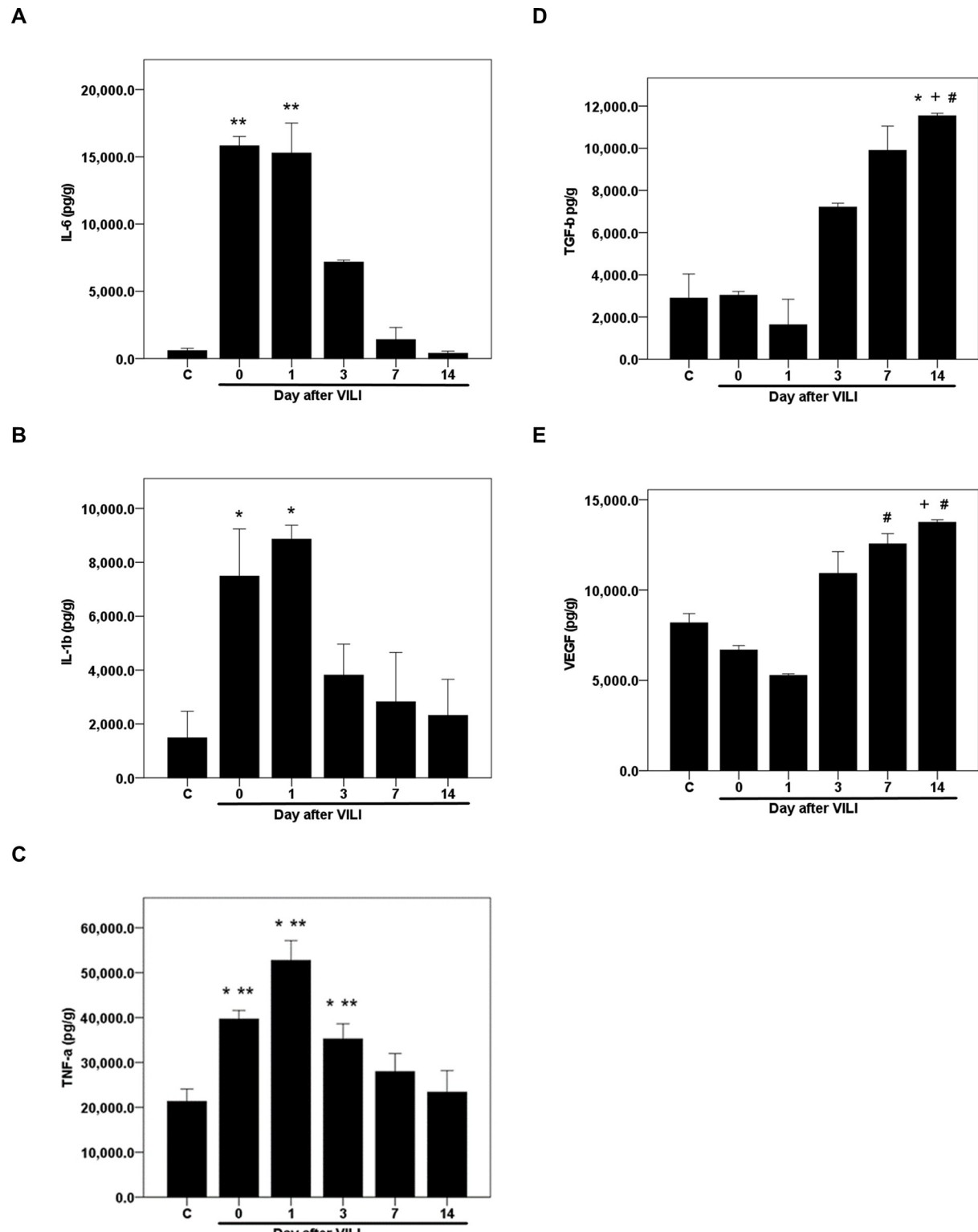

**Fig 3. Time courses of pulmonary inflammatory cytokine and growth factor levels following VILI.** ELISA for inflammatory cytokines and growth factors in the lung tissue, (A) interleukin-6 (IL-6), (B) interleukin-1 beta (IL-1β), (C) tumor necrosis factor-alpha (TNF-α), (D) tumor growth factor-beta (TGF-β), and (E) vascular endothelial growth factor (VEGF). Mice were sacrificed immediately after VILI (day 0) or on days 1, 3, 7, or 14 after VILI. Statistical analysis was performed using parametric (one-way ANOVA with Bonferroni's or Dunnett's post hoc test) or nonparametric (Kruskal-Wallis test with Bonferroni correction) tests. Data are expressed as means ± SD; n = 4–6 per group, $p < 0.05$ for * vs. normal control (C), + vs. day 0, # vs. day 1 and ** vs. day 14.

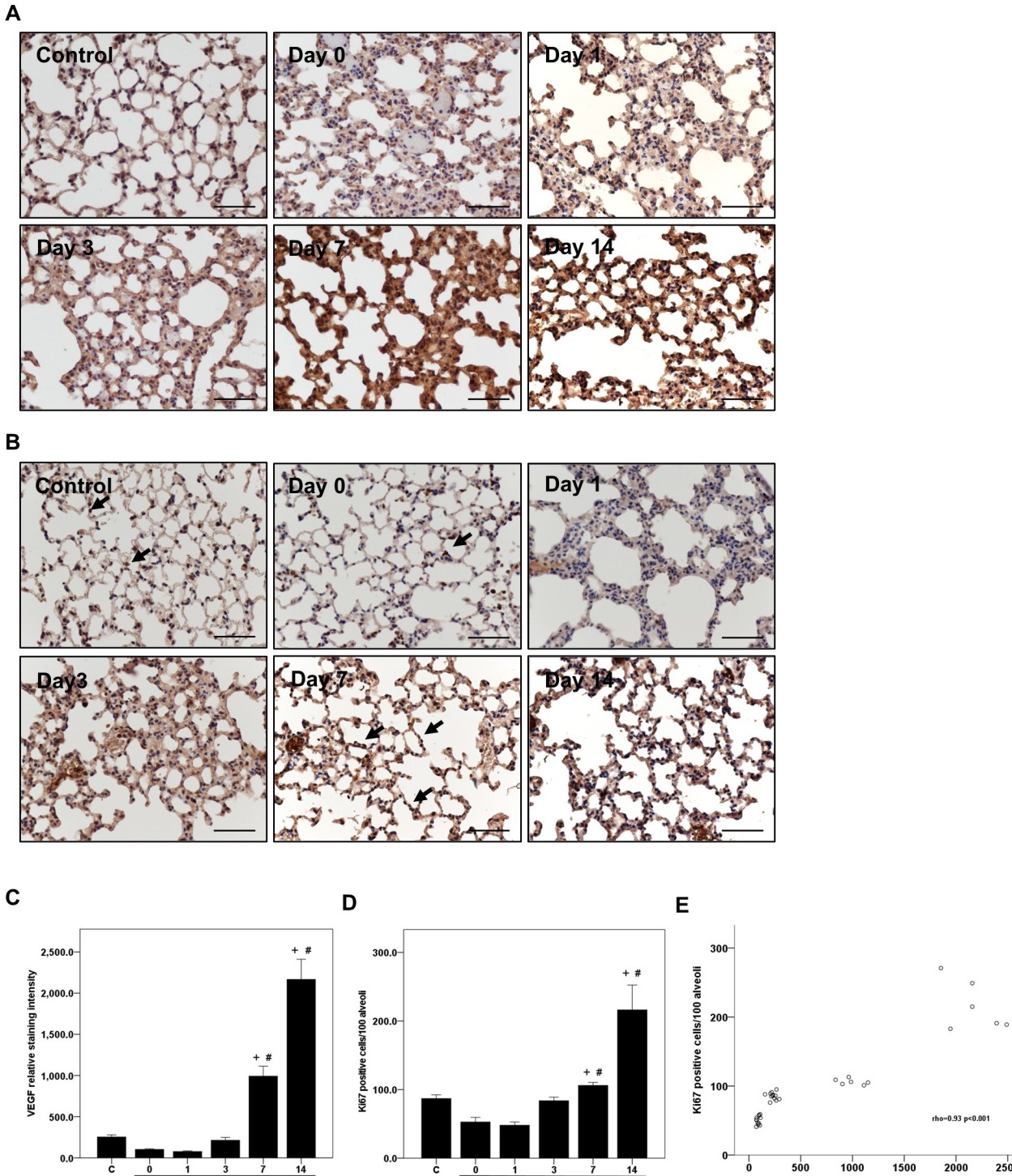

**Fig 4. Immunocytochemistry for VEGF and Ki67 in lung sections of mice following VILI.** C57BL/6 mice received 20 ng LPS intraventricularly and HTV ventilation to produce a two-hit VILI. The mice were sacrificed immediately after VILI (day 0) or on day 1, 3, 7, or 14 after VILI. (A) Immunocytochemistry (IHC) for VEGF in mouse lung sections. (B) IHC for Ki67-positive cells of the alveolar epithelium in mouse lung sections. The closed arrow indicates Ki67-positive cells in the alveolar epithelium. (C-D) Histometric analysis of the time courses of VEGF expression and the number of Ki67-positive cells in the lung alveolar epithelium. Scale bar = 200 μm. Statistical analysis was performed using the Kruskal-Wallis test with Bonferroni correction. Data are expressed as means ± SD; n = 6 per group, $p < 0.05$ [+] vs. day 0 and [#] vs. day 1; normal control (C). (E) Correlation between the VEGF relative staining intensity and the number of Ki67-positive cells per 100 alveoli (Spearmen's correlation coefficient rho = 0.93, $p < 0.001$).

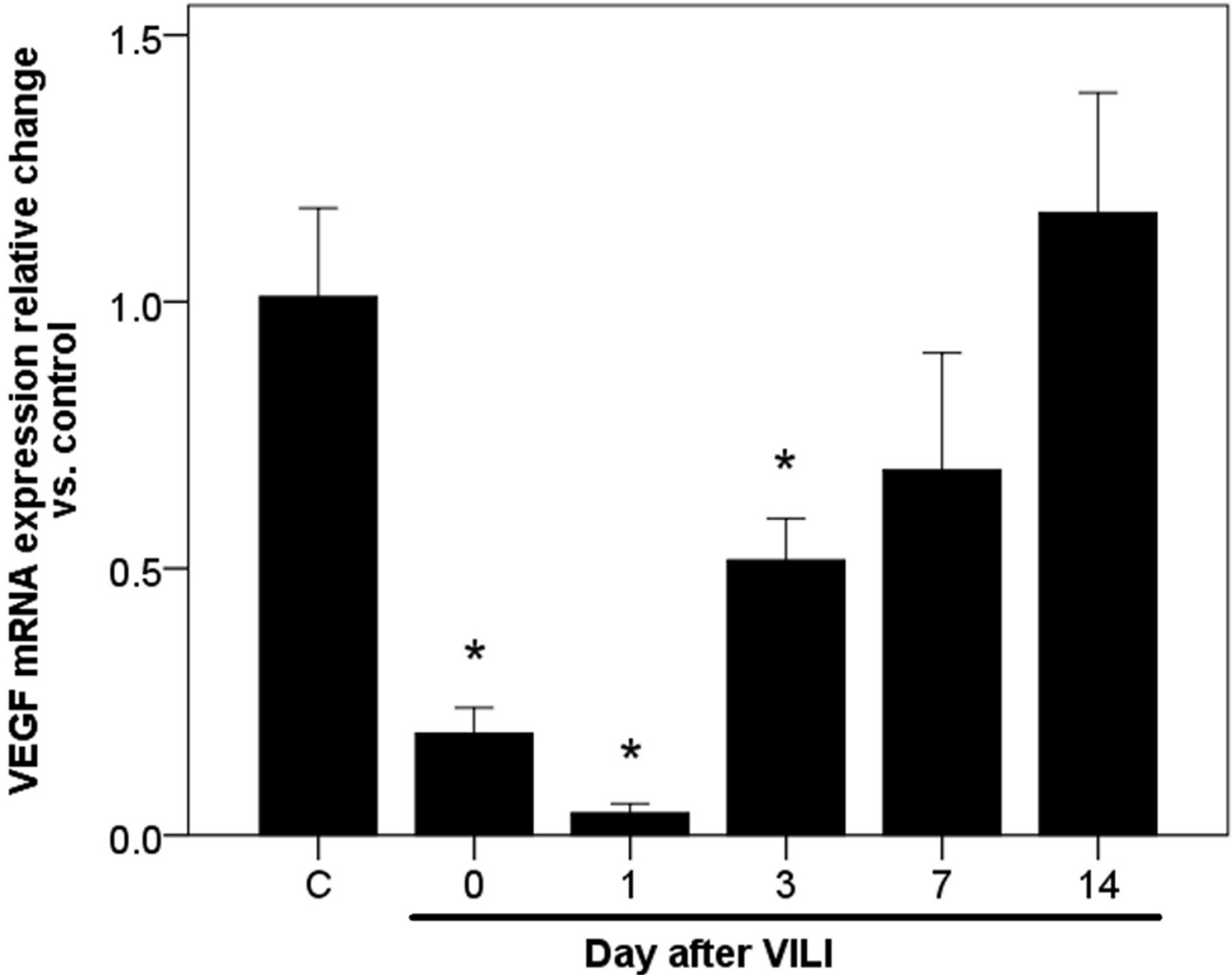

**Fig 5. Time course of VEGF mRNA expression in lung-recruited monocytes following VILI.** mRNA expression was presented as the relative change versus untreated control. Statistical analysis was performed using ANOVA. Data are expressed as means ± SD; n = 4–5 per group, $^*p < 0.05$, vs. normal control (C).

depleted monocytes from mice using liposomal clodronates from day 3 after VILI, when the pulmonary VEGF began to restore. The depletion significantly suppressed the pulmonary VEGF restoration, which declined by 24% on day 7 (12556 ± 567.5 vs. 9544.4 ± 1069.9 pg/g; $p = 0.002$, Fig 6) and 47% on day 14 (13755 ± 138.5 vs. 7301.9 ± 1602.8 pg/g; $p = 0.002$, Fig 6).

## Discussion

We used the two-hit model of lung injury to investigate the roles of VEGF and lung-recruited monocytes in post-VILI lung repair. The main findings of this study are as follows. First, during recovery from VILI, a majority of lung-recruited monocytes are Ly6C$^{+low}$. Second, the expression of growth factors, such as VEGF and TGF-β, in injured lungs increases progressively during recovery from VILI, while inflammation fades, as determined from the decreasing levels of IL-6, TNF-α, and IL-1β. Moreover, VEGF expression in lung tissue positively

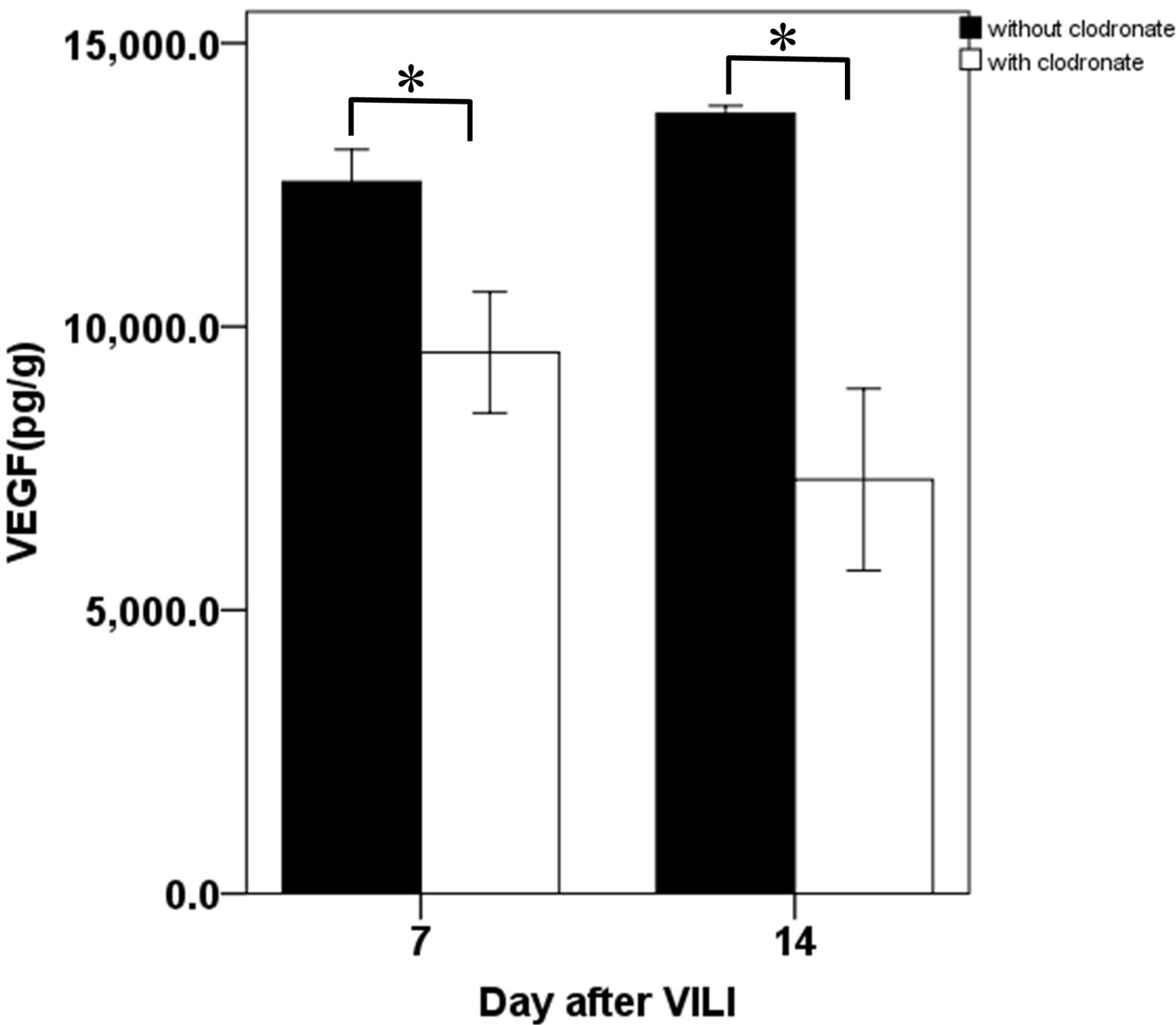

**Fig 6. Pulmonary VEGF protein levels in mice without and with depletion of monocytes on days 7 and 14 after VILI.** Mice treated with clodronate for depleting pulmonary monocytes underwent the same protocol as the two-hit VILI. The mice sacrificed on day 7 after VILI were injected with a single 200 uL dose of Clophosome-A intravenously on day 3 after VILI. The mice sacrificed on day 14 after VILI received the same treatment of Clophosome-A on day 3 and reinjection of 100 uL on days 7 and 11 after VILI. VEGF protein concentration in lung tissue was determined by ELISA. Data are expressed as mean ± SD; n = 6 per group, *p = 0.002, compared between without and with clodronate treatment groups using Mann-Whitney U Test.

correlated with the expression of Ki67, a cell proliferation marker, in alveolar epithelial cells. Third, monocytes are a cellular source of VEGF and the depletion of monocytes suppresses the post-VILI restoration of pulmonary VEGF.

In patients on ventilator support, bacterial infection, as one of the activators, can activate circulating inflammatory cells. In the current study, a two-hit mouse model was used to mimic the clinical scenario, a subclinical dose of LPS was used to prime circulating inflammatory cells but not damage the alveolar epithelial barrier, and a setting of HTV resulting in VILI [17,

21, 22]. There remains a substantial difference between live bacterial infection and LPS treatment in terms of cytokine expression profiles [23, 24]. The overexpression of inflammatory cytokines in mice exposed to either LPS or peritoneal contamination and infection recovers after 72 h. However, cecal ligation and puncture treatment induce comparatively fewer effects with a more protracted inflammation course. Considering the experimental complexity and difficulty, the LPS model is more suitable and proper for investigating acute inflammation and subsequent tissue repair, as in our study.

This study found that Ly6C$^{+low}$ monocytes are recruited to the lungs during recovery from VILI. The roles of monocytes in inflammatory tissue vary depending on their surface Gr1 (Ly6C) expression. The Gr1$^+$ (Ly6C$^+$) subset of monocytes recruited to inflamed tissues participates in inflammatory responses [10, 25]. In contrast, the Gr1$^-$ (Ly6C$^-$) subset of monocytes, which surveys the vasculature, is involved in tissue repair [13, 14]. Nahrendorf *et al.* [15] discovered the biphasic responses and dual roles of monocytes in myocardial injury, in which Ly6C$^{+high}$ monocytes, which dominate during the early stage after myocardial infarction, exhibit inflammatory functions, and Ly6C$^{+low}$ monocytes, which become more abundant during the healing phase, attenuate inflammation. In ALI, pulmonary macrophages are present as diverse populations that mediate acute lung inflammation and resolution separately [26]. Pulmonary macrophages are derived from circulating monocytes [27]. Circulating monocytes primed by endotoxemia and recruited to the lungs can also predispose the lungs toward acute injury [21, 25]. In VILI, mechanical stress of HTV ventilation contributes to injury by upregulating intrapulmonary cytokines expression [28]. The lung-recruited monocytes, which mainly express Gr1$^{high}$, can be further activated during HTV ventilation and promote the progression of VILI [17, 29]. As in myocardial injury, we found that the biphasic monocyte response also exists in mouse VILI, where Ly6C$^{+high}$ monocytes dominate during the initiation of VILI, and Ly6C$^{+low}$ monocytes become more abundant during recovery from VILI. These findings suggest that Ly6C$^{+low}$ monocytes may play a crucial role in resolving VILI.

During recovery from the VILI insult, the temporal expression of VEGF and TGF-β follows an opposite trend to that of inflammatory cytokines. The concentration of VEGF and TGF-β in the injured lungs increased progressively and was significantly higher than the baseline and those in the first 24 h after injury. To the best of our knowledge, most of the previous studies focused on the expression and contribution of inflammatory cytokines during the early inflammatory phase of lung injury. Few of the previous studies addressed the subsequent temporal expression pattern and explored the roles of inflammatory cytokines and growth factors during lung healing. However, it is of paramount importance to understand the temporal expression pattern of inflammatory cytokines and growth factors during the recovery phase after lung injury, which helps in elucidating the mechanism underlying lung repair. TNF-α, IL-1β, and IL-6, the proinflammatory cytokines, are known to be involved in VILI [30]. In the early phase of VILI, Wilson *et al.* [17] found that TNF-α expression was rapidly upregulated in bronchoalveolar lavage fluid (BALF) after 2 h of HTV. Our previous study based on a two-hit VILI mouse model also showed that the elevated IL-6 and TNF-α concentrations in BALF and serum peaked at 4–6 h after HTV ventilation [18]. In acid-induced ALI, Patel *et al.* [31] found that the TNF-α and IL-6 in BALF peaked between days 1 and 3, followed by a return to normal levels over days 5 to 10. All the above studies explored the time course of inflammatory cytokines in BALF but not in injured lung tissue. To understand the expression of inflammatory cytokines and growth factors in injured lungs, we investigated the temporal expression patterns of both inflammatory cytokines and growth factors in lung tissue lysates of mice subjected to VILI. We found that VILI-induced elevation in IL-6, IL-1β, and TNF-α levels in the lung tissue lysates peaked between days 0 and 1 and returned to the baseline on day 7 after VILI. In contrast, the concentrations of VEGF and TGF-β in lung tissue lysates decreased

immediately after VILI, but progressively increased and remained at a higher level than the baseline on day 14 after VILI. The findings of VEGF expression in lung tissue following VILI are different from those of VEGF expression in BALF and serum reported in our previous study [18]. However, these findings are consistent with the results obtained in the oleic-acid-induced ALI of Guo et al. [32], where the expression of VEGF increased in BALF and serum but decreased in the lung tissue [32]. Our findings can be explained by a proposed mechanism of VEGF downregulation in lung tissue during ALI, in which alveolar epithelial cells and endothelial cells, the constitutive source of VEGF in the lungs, are reduced in number due to direct lung injury [33, 34].

The augmented expression of VEGF and TGF-β in the injured lungs highlighted the roles played by VEGF and TGF-β in post-VILI lung repair. Proteins belonging to epidermal growth factor and fibroblast growth factor families are involved in repairing injured lung epithelium [4]. VEGF, besides its well-known angiogenetic properties, acts as a potent lung epithelial mitogen involved in resolving lung injury [35–38]. Studies have suggested that VEGF plays an important role in recovery from ALI, but the findings remain debatable [33, 39–42]. The ambiguous results may be due to the variation in VEGF obtained from different sites at different time points after ALI. This implies that the effects of VEGF vary with the affected cells in the lungs and depend on the time points evaluated after ALI. VEGF receptor signaling is required for the maintenance of alveolar structures in normal rat lungs, and also regulates the proliferation and apoptosis of injured alveolar epithelial cells in an autocrine or paracrine manner [7, 43, 44]. Acute respiratory distress syndrome (ARDS) is the most severe form of ALI. Medford et al. [45] found that VEGF expression was significantly upregulated in late ARDS after day 7 compared to both normal subjects as well as early ARDS within 48 hours. Pneumonia is one of the direct causes of ARDS. Strouvalis et al. [46] showed an association between increased concentration of circulating VEGF and resolution of ventilator-associated pneumonia (VAP). The findings of their animal-based experiments provide evidence that VEGF plays a role in the resolution of lung injury starting from 5 days post- lung injury. In the present study, we analyzed the expression of VEGF and the number of Ki67-positive cells in the alveolar epithelium of lung sections and found a strong positive correlation between VEGF expression and the proliferation of alveolar epithelial cells during VILI resolution. The results suggest that VEGF may contribute to alveolar reconstruction in the injured lungs during healing from VILI. Taken together, these findings suggest a dual role for VEGF in VILI. The increased concentration of VEGF in BALF and serum contribute to progressive lung injury in the early phase of VILI, but the elevated expression of VEGF in the lung tissue facilitates lung repair during the late recovery phase. Like VEGF, the contribution of TGF-β in lung repair is both complex and controversial but is more suggestive of failed repair and pulmonary fibrosis [47]. Our work on TGF-β is ongoing, and its involvement is yet to be fully elucidated.

As previously mentioned, alveolar epithelial cells, as a primary source of pulmonary VEGF, are destroyed during the initial phase of ALI. This results in decreased expression of VEGF in the injured lung tissue [20, 33]. In the recovery period following VILI, we found that VEGF increased progressively, and lung-recruited Ly6C$^{+low}$ monocytes showed a similar trend in VEGF expression. Monocytes and macrophages belong to the mononuclear phagocyte system and circulating monocytes are the precursors of pulmonary macrophages [48, 49]. During inflammation, Ly6C$^{+high}$ monocytes are rapidly recruited to sites of injury where they extravasate and differentiate into macrophages [9]. In injured tissue, inflammation and phagocytosis of apoptotic cells prompt macrophages to synthesize and release VEGF, which is crucial for tissue repair after injury [38, 50, 51]. Steinberg et al. [52] showed that in ARDS, a progressive increase in alveolar macrophages is associated with patient survival, and Eric D et al. [53] discovered that clinical outcomes depend on distinct alveolar macrophage transcriptional

programs. David *et al.* [39] reported that VEGF levels in alveolar macrophages and epithelial lining fluid (ELF) from patients with ARDS were significantly less than those from at-risk subjects, and the increased VEGF levels in the ELF at day 4 were associated with recovery. These findings suggest that macrophages could be an essential source of VEGF and contribute to tissue repair in the injured lungs. Monocytes, like macrophages, can also produce VEGF [54]. In this study, we found that the temporal expression of pulmonary VEGF proteins was consistent with that of VEGF mRNA in lung-recruited monocytes, and that depletion of monocytes suppressed the restoration of pulmonary VEGF during resolution of VILI. These findings suggest that lung-recruited monocytes, mainly expressing Ly6C$^{+low}$, are a cellular source of pulmonary VEGF in the injured lungs and may drive healing process via secreting VEGF to promote epithelial repair.

There are several notable limitations to the present study. First, some of the experimental mice have died before being sacrificed. The cause of death may have been excessive inflammation and lung injury. This suggests that the concentrations of inflammatory cytokines in the injured lung tissue may have been underestimated. Second, there is a lack of direct evidence of a causal relationship between increased pulmonary VEGF and post-VILI proliferation of alveolar epithelial cells. We did not directly suppress the expression of pulmonary VEGF to examine the influence on the number of Ki67-positive cells in the alveolar epithelium during recovery from VILI. Since VEGF plays a crucial role in angiogenesis, systemic VEGF inhibition may produce unpredictable confounding effects on the estimation. In addition, there is no effective and selective method to precisely silence or inhibit the expression of VEGF in the lungs. Thus, further studies are required to resolve this issue. Third, the subset of monocytes expressing Ly6C$^{+high}$ or Ly6C$^{+low}$ and contributing to the restoration of pulmonary VEGF in injured lungs was not characterized. Due to the lack of an available method to deplete monocytes selectively *in vivo*, it is difficult to elucidate the differential effects of monocytes with different phenotypes on the restoration of pulmonary VEGF. However, our findings provide credible evidence that lung-recruited monocytes expressing Ly6C$^{+low}$ are the predominant cells involved in resolving lung injury.

## Conclusion

During recovery from VILI, the time courses of pulmonary growth factors are different from those of inflammatory cytokines. Pulmonary VEGF level remains higher than the baseline and is associated with pulmonary epithelial proliferation. During VILI, monocytes participate in two distinct phases. Lung-recruited Ly6C$^{+high}$ monocytes dominate during the initiation of VILI, but Ly6C$^{+low}$ monocytes become more abundant during the recovery phase following VILI. Monocytes are a cellular source of VEGF and contribute to post-VILI restoration of pulmonary VEGF. Thus, lung-recruited monocytes and pulmonary VEGF may play crucial roles in post-VILI lung repair.

## Supporting information

**S1 Fig. Flow chart depicting animal experiment design.** The numbers of animals used in the different experimental groups and the numbers of animals died during mechanical ventilation and recovery.
(TIF)

**S1 Text. Detailed illustrations of the statistical analysis.**
(DOCX)

## Author Contributions

**Conceptualization:** Chin-Kuo Lin, Chung-Sheng Shi.

**Data curation:** Chin-Kuo Lin, Tzu-Hsiung Huang.

**Formal analysis:** Chin-Kuo Lin.

**Funding acquisition:** Chin-Kuo Lin.

**Investigation:** Chin-Kuo Lin, Tzu-Hsiung Huang.

**Methodology:** Chin-Kuo Lin.

**Resources:** Tzu-Hsiung Huang.

**Supervision:** Cheng-Ta Yang, Chung-Sheng Shi.

**Validation:** Chin-Kuo Lin, Chung-Sheng Shi.

**Visualization:** Tzu-Hsiung Huang, Cheng-Ta Yang, Chung-Sheng Shi.

**Writing – original draft:** Chin-Kuo Lin.

**Writing – review & editing:** Cheng-Ta Yang, Chung-Sheng Shi.

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
