## [Editor Report · Decision Letter 0]

18 Sep 2020

PONE-D-20-27843

Roles of lung-recruited monocytes and pulmonary vascular endothelial growth factor (VEGF) in resolving ventilator-induced lung injury (VILI)

PLOS ONE

Dear Dr. Shi,

Thank you for submitting your manuscript to PLOS ONE. After careful consideration, we feel that it has merit but does not fully meet PLOS ONE’s publication criteria as it currently stands. Therefore, we invite you to submit a revised version of the manuscript that addresses the points raised during the review process.

Before inviting expert reviewers, I checked your manuscript and noticed that there are neither details about the number of animals per group and experiments nor about the number of relications of experiments. Please include these data in the Material & Methods section as well as in the figure legends. In addition, you may wish to include the detailed descriptions of the methods, which you give in the supplement, directly into the main text.

Please revise and re-submit as soon as possible so that I can invite the reviewers.

We look forward to receiving your revised manuscript.

Kind regards,

Heinz Fehrenbach

Academic Editor

PLOS ONE

Journal Requirements:

3. Please provide the specific sequences of th eprimers used in the PCR analysis.

4. At this time, we ask that you please provide scale bars on the microscopy images presented in Figures 1 and 4, and refer to the scale bar in the corresponding Figure legend.

5. Please provide the product numbers and any lot numbers of the antibodies purchased for immunohistochemistry in your study.”

6. At this time, we request that you  please report additional details in your Methods section regarding animal care, as per our editorial guidelines:

(a) Please state the number of mice used in the study

(b) Please provide details of animal welfare (e.g., shelter, food, water, environmental enrichment)

(c) Please describe the post-operative care received by the animals, including the frequency of monitoring and the criteria used to assess animal health and well-being.

Thank you for your attention to these requests.

7. To comply with PLOS ONE submissions requirements, in your Methods section, please provide additional information regarding the experiments involving animals and ensure you have included details of the type and dosage of anesthesia, both during ventilation and for sacrifice.

8. Please ensure your methods are described in sufficient detail for others to replicate the study. Specifically, please provide the following:

i) Please provide the names of the ELISA kits used to quantify pulmonary growth factors and cytokines.

ii) Please include the Methods described in the appendix in the main manuscript body."

9. To comply with PLOS ONE submission guidelines, in your Methods section, please provide additional information regarding your statistical analyses. For more information on PLOS ONE's expectations for statistical reporting, please see https://journals.plos.org/plosone/s/submission-guidelines.#loc-statistical-reporting

10. PLOS requires an ORCID iD for the corresponding author in Editorial Manager on papers submitted after December 6th, 2016. Please ensure that you have an ORCID iD and that it is validated in Editorial Manager. To do this, go to ‘Update my Information’ (in the upper left-hand corner of the main menu), and click on the Fetch/Validate link next to the ORCID field. This will take you to the ORCID site and allow you to create a new iD or authenticate a pre-existing iD in Editorial Manager. Please see the following video for instructions on linking an ORCID iD to your Editorial Manager account: https://www.youtube.com/watch?v=_xcclfuvtxQ

Additional Editor Comments:

Before inviting expert reviewers, I checked your manuscript and noticed that there are neither details about the number of animals per group and experiments nor about the number of relications of experiments. Please include these data in the Material & Methods section as well as in the figure legends. In addition, you may wish to include the detailed descriptions of the methods, which you give in the supplement, directly into the main text.

Please revise and re-submit as soon as possible so that I can invite the reviewers.
---

## [Author Response · Author response to Decision Letter 0]

22 Nov 2020

Dear Dr. Heinz Fehrenbach,

Thank you for your letter and the opportunity to revise our paper in PLOS ONE. We appreciate your insightful comments on revising the Materials & Methods of the paper.

I have included the comments immediately after this letter and responded to them individually, indicating exactly how we addressed each concern or problem and describing the changes we have made. The revisions have been approved by all four authors, and I have again been chosen as the corresponding author. The changes are marked in red in the paper as you requested, and the marked and unmarked versions of the revised manuscript have been uploaded to the online submission system.

In response to your comments on the Materials & Methods section, we have described the number of animals per group for each experiment and the number of replications in each experiment. In addition, we have included in the main text detailed descriptions of the methods, which were originally presented in the supplementary information.

We hope the revised manuscript will better suit PLOS ONE but are happy to consider further revisions, and we thank you for your continued interest in our research.

Sincerely,

Chung-Sheng Shi

Graduate Institute of Clinical Medicine Sciences, College of Medicine, Chang Gung University,

 

Editor comments, Journal Requirements, Author Responses and Manuscript Changes

Editor Comments:

Before inviting expert reviewers, I checked your manuscript and noticed that there are neither details about the number of animals per group and experiments nor about the number of relications of experiments. Please include these data in the Materials & Methods section as well as in the figure legends. In addition, you may wish to include detailed descriptions of the methods, which you give in the supplement, directly into the main text.

Response: 

We have rewritten and detailed the number of animals per group in each experiment and the number of replications for each experiment in the Materials and Methods section. In addition, we have moved the detailed descriptions of the methods initially presented in the supplementary information directly into the Materials and Methods section of the main text.

Journal Requirements

1. Please ensure that your manuscript meets PLOS ONE's style requirements, including those for file naming. The PLOS ONE style templates can be found at https://journals.plos.org/plosone/s/file?id=wjVg/PLOSOne_formatting_sample_main_body.pdf and https://journals.plos.org/plosone/s/file?id=ba62/PLOSOne_formatting_sample_title_authors_affiliations.pdf. 

Response:

We have reformatted our manuscript and file names according to the journal style requirements.

2. We suggest you thoroughly copyedit your manuscript for language usage, spelling, and grammar. If you do not know anyone who can help you do this, you may wish to consider employing a professional scientific editing service. While you may use any professional scientific editing service of your choice, PLOS has partnered with both American Journal Experts (AJE) and Editage to provide discounted services to PLOS authors. Both organizations have experience helping authors meet PLOS guidelines and can provide language editing, translation, manuscript formatting, and figure formatting to ensure that your manuscript meets our submission guidelines. To take advantage of our partnership with AJE, visit the AJE website (http://learn.aje.com/plos/) for a 15% discount off AJE services. To take advantage of our partnership with Editage, visit the Editage website (www.editage.com) and enter the referral code PLOSEDIT for a 15% discount off Editage services. If the PLOS editorial team finds any language issues in text that either AJE or Editage has edited, the service provider will re-edit the text for free.

Response:

We have had our manuscript professionally edited by Editage.

We have uploaded a copy with red text highlighting the changes and a clean copy as the new manuscript file.

3. Please provide the specific sequences of the primers used in the PCR analysis.

Response:

The primer sequences used for RT-PCR analysis of VEGF mRNA have been included in the Materials and Methods section (page 12, line 201). The primer sequences of VEGF and GADPH (endogenous control) are as follows: 

VEGF forward: 5’-ATCATGCGGATCAAACCTCACCA-3’; 

VEGF reverse: 5’-TCACCGCCTTGGCTTGTCACAT-3’; 

GAPDH forward 5’-CTCACTGGCATGGCCTTCCGTGT-3’, 

GAPDH reverse: 5’-GATGTCATCATATTTGGCAGGT-3’.

4. At this time, we ask that you please provide scale bars on the microscopy images presented in Figures 1 and 4, and refer to the scale bar in the corresponding Figure legend.

Response:

We have added scale bars to the microscopy images in Figures 1 and 4 (page 15, line 251 and page 19, line 342). The scale bars represent 200 �m, as described in the corresponding figure legends. 

5. Please provide the product numbers and any lot numbers of the antibodies purchased for immunohistochemistry in your study.”

Response:

We have edited the Materials and Methods section and detailed the product numbers and lot numbers of the antibodies used for immunohistochemistry analysis in our study (page 8, line 122).

6. At this time, we request that you please report additional details in your Methods section regarding animal care, as per our editorial guidelines:

(a) Please state the number of mice used in the study

(b) Please provide details of animal welfare (e.g., shelter, food, water, environmental enrichment); 

(c) Please describe the postoperative care received by the animals, including the frequency of monitoring and the criteria used to assess animal health and well-being.

Thank you for your attention to these requests.

Response:

(a) We have described the number of animals per group for each experiment and the number of replications for each experiment in the Materials and Methods section (page 7, line 112; page 9, line 136; page 10, line 156; page 10, line 167 and page 13, line 211).

(b) and (c)

We have edited the Methods section and described the animal welfare (page 6, line 80) and post-treatment (page 7, line 107). 

For the animal welfare, all experimental protocols using animals were performed following the Guide to the Care and Use of Experimental Animals and approved by the Institutional Animal Care and Use Committee of Chang Gung Memorial Hospital, Chiayi, Taiwan (Approval number: 2012102301 (September 21, 2013)). All animal experiments were performed in the Animal Center of Chang Gung Memorial Hospital at Chiayi, and mice were housed following IACUC, national, and international animal welfare protocols. The center introduced the AAALAC certification system in 2013 and had obtained AAALAC certification in July 2014 (No: TTQ08384). To optimize environmental conditions for reducing mouse stress on animal welfare, animals were housed in a room with constant temperature (20–25°C) and humidity (40–60%) under a 12‐h light/dark cycle, mouse cages with enrichment toys were limited to 5 mice per cage, animals were given free access to food and water, and the center was maintained in low noises and vibrations.

For the postoperative care, the extubated mice received post-treatment care following the IACUC guidelines. We checked the appearance of mice every day and measured the body weight of mice twice/week. Besides, a full‐time veterinarian would take care of animal health and well-being until they were sacrificed with an overdose of anesthesia on days 1, 3, 7, or 14 after VILI.

7. To comply with PLOS ONE submissions requirements, in your Methods section, please provide additional information regarding the experiments involving animals and ensure you have included details of the type and dosage of anesthesia, both during ventilation and for sacrifice.

Response:

We have detailed the anesthesia of the animals, including the drug, type, and dosage of anesthesia (page 7, line 97), during ventilation (page 7, line 103), and for sacrifice (page 7, line 105). Briefly, mice were anesthetized using intraperitoneal Zoletil 50 (80 mg/kg; Tiletamine-Zolazepam, Virbac, Carros CEDEX, France) and received a single intravenous tail vein injection of 20 ng LPS (O111B4; Sigma-Aldrich, St Louis, MO, USA) immediately before HTV ventilation. The mice were intubated and ventilated with an SAR-830 series small animal ventilator with multi-animal setups (CWE Inc., Ardmore, PA, USA) under room air and general anesthesia with regular intraperitoneal injections of Zoletil 50 (10 mg/kg/h) for 4 h. After ventilation, the mice were sacrificed immediately (day 0) or extubated after spontaneous breath resumption and sacrificed on day 1, 3, 7, or 14 after VILI by an overdose of anesthesia (Zoletil 50, 50 mg/kg). 

8. Please ensure that your methods are described in sufficient detail for others to replicate the study. Specifically, please provide the following:

i) Please provide the names of the ELISA kits used to quantify pulmonary growth factors and cytokines.

ii) Please include the Methods described in the appendix of the main manuscript body."

Response:

We have edited the Materials and Methods section and described the names and lot numbers of the ELISA kits used in the quantification of pulmonary growth factors and cytokines (page 10, line 161). In addition, we have moved the detailed descriptions of the methods initially presented in the supplement, directly into the Materials and Methods section of the main text.

9. To comply with PLOS ONE submission guidelines, in your Methods section, please provide additional information regarding your statistical analyses. For more information on PLOS ONE's expectations for statistical reporting, please see https://journals.plos.org/plosone/s/submission-guidelines.#loc-statistical-reporting

Response:

We have carefully read PLOS ONE's expectations for statistical reporting and SAMPL guidelines for general guidance on statistical reporting and ensured the description of the statistical analysis section complies with the essential principle. For instance, the name and version of statistical software, the form of summarized data, total sample and group sizes for each analysis, the method used for each analysis, and the alpha level for defining statistical significance has been detailed in the Methods section (page 13, line 222). Besides, detailed illustrations of the statistical analysis are provided in the supporting information S1 Text.

10. PLOS requires an ORCID iD for the corresponding author in the Editorial Manager on papers submitted after December 6th, 2016. Please ensure that you have an ORCID iD and that it is validated in the Editorial Manager. To do this, go to update my information (in the upper left-hand corner of the main menu), and click on the Fetch/Validate link next to the ORCID field. This will take you to the ORCID site and allow you to create a new iD or authenticate a pre-existing iD in Editorial Manager. Please see the following video for instructions on linking an ORCID iD to your editorial manager account: https://www.youtube.com/watch?v=_xcclfuvtxQ. 

Response:

The ORCID iD of the corresponding author, Chung-Sheng Shi, has been edited, linked, and validated in the online submission system of the Editorial Manager.

---

## [Decision Letter · Decision Letter 1]

5 Jan 2021

PONE-D-20-27843R1

Roles of lung-recruited monocytes and pulmonary vascular endothelial growth factor (VEGF) in resolving ventilator-induced lung injury (VILI)

PLOS ONE

Dear Dr. Shi,

Thank you for submitting your manuscript to PLOS ONE. After careful consideration, we feel that it has merit but does not fully meet PLOS ONE’s publication criteria as it currently stands. Therefore, we invite you to submit a revised version of the manuscript that addresses the points raised during the review process.

Two expert reviewers from the field raised a number of points that have to be addressed when revising the manuscript. In particular, all details such as animal numbers and handling, outcome of the experiments, statistical tests used as well as methodological issues have to be included so that the study can be fully appreciated by our readers. Please submit a detailed point-by-point response to all comments along with your revised manuscript. In addition, I suggest that you get a native speaker involved in editing the English language.

We look forward to receiving your revised manuscript.

Kind regards,

Heinz Fehrenbach

Academic Editor

PLOS ONE

Reviewers' comments:

Reviewer's Responses to Questions

**Comments to the Author**

1. If the authors have adequately addressed your comments raised in a previous round of review and you feel that this manuscript is now acceptable for publication, you may indicate that here to bypass the “Comments to the Author” section, enter your conflict of interest statement in the “Confidential to Editor” section, and submit your "Accept" recommendation.

Reviewer #1: All comments have been addressed

Reviewer #2: All comments have been addressed

2. Is the manuscript technically sound, and do the data support the conclusions?

Reviewer #1: (No Response)

Reviewer #2: Partly

3. Has the statistical analysis been performed appropriately and rigorously? 

Reviewer #1: (No Response)

Reviewer #2: I Don't Know

4. Have the authors made all data underlying the findings in their manuscript fully available?

Reviewer #1: Yes

Reviewer #2: No

5. Is the manuscript presented in an intelligible fashion and written in standard English?

Reviewer #1: Yes

Reviewer #2: Yes

6. Review Comments to the Author

Reviewer #1: The resubmitted manuscript entitled „Roles of lung-recruited monocytes and pulmonary vascular endothelial growth factor (VEGF) in resolving ventilator-induced lung injury (VILI)” by Chung-Sheng Shi et al. describes the role of monocytes and the release of VEGF in a model of VILI. The authors described

All points of the reviewer are properly edited the statistics and described the different statistical models in supplement. All numbers of the experiments were mentioned and statistical variances shown as mean � SD. The text body was well corrected and most of the typos removed. There are some miner points I would like to address.

.

Minor points:

1. In figure 4B, day 1 the authors should check if the magnification is really the same as in the other histological images. It seems to be a higher magnification. Please clarify.

2. The correlation between the pulmonary VEGF and alveolar epithelial barrier one side and the role of monocytes on the other side are two possible explanations for the changes of VEGF during VILI. However, the authors work out the differences more detailed in the discussion.

3. In the discussion section (line 378-381) the authors compared the bacterial infection in ventilated patients with their two-hit model of LPS and ventilation. The author should clarify that there is a substantial difference between bacterial infection and LPS treatment in terms of cytokine profiles

Reviewer #2: General comments.

Please, clearly decribe how many animals were used in each Group. How many animals entered a Group, how many survived VILI, how many were sacrificed early, etc. Discuss, speculate how this may have impacted on the results? Did animals with an excessive Inflammation die early? Did you check?

Consider a figure, flow chart.

Please, clearly decribe the ventilation used to induce VILI.

Please, clearly describe the anima handling after the experiment. Did the mice receive pain medication, etc.

The statistics used may not be appropriate. You used parametric tests despite the low n. The use of non-parametric test would be more cautious. Use non-parametric tests or explain. Did you check for normal distribution in all groups?

Avoid nonsensical phrases: The results reveal, etc.

Always state the measurement unit.

Line 79

Animals and two-hit model of VILI

The description of the animal model is insufficient.

Clearly describe each hit of the two hit model, e.g.:

The first hit consisted of ……..

The second hit ……..

Then, give a detailed report of the experimental procedure in the correct chronological order.

i. anesthesia

ii. LPS-injection

iii. Intubation

iv. Ventilation

v. Extubation

Specify, the ventilation used to induce VILI besides tidal volume and respiratory rate further: inspiration time, I:E, resulting peak pressure and duration of ventilation!

Whan did you stop ventilation. How and when as a mouse extubated? Did you extubate right from VILI ventilation? What kind of ventilation did you used to wean anesthesia.

Line 92:

….water, and the center was maintained in low noises and vibrations.

Sounds awkward. Rephrase

Line 112:

The number of animals used per group was as follows: normal control (n = 6), day 0 (n = 6 ), day 1 (n = 6), day 3 (n = 6), day 7 (n = 12, 6 of them for the study of depletion of pulmonary monocytes), and day 14 (n = 12, 6 for the study of depletion of pulmonary monocytes).

This sounds to good. Not one animal died? How many animals did you loose during induction of VILI. How many animals entered each group and how many remained over the following days? What was the selection process to euthanize an animal prematurely?

It needs to be discussed in the discussion section in how far that may have altered the results.

Line 117:

How were the lungs inflated – manually? With a defined pressure/volume?

Line 215:

How were the mice sedated for this injection. Did you need to ventilate them?

Line 222:

You used parametric tests despite the small number of animals. Did you check for normal distribution? Non-parametric tests and Spearman correlation may be the more conservative strategy.

Line 245:

C57BL/6 mice received i.v. with 20 ng of LPS and high-stretch mechanical ventilation that produced two-hit VILI.

Sentence does not make sense. Rephrase.

Line 253:

….; n = 3-5, ….

What does that n mean? It does not match the group size given in the method section?

This refers to all other n = ..... in the figure descriptions.

Line 262:

Avoid nonsencical phrases like: The results revealed ……

Line 281:

..SD; n = 4-5,

What is this n?

General comment:

What is a background level of something in mice not held under experimental conditions?

If you mean the first measured level before the intervention you may want to use the term baseline. Do you mean the level in the control animals at that specific time point?

Consider using concentration instead of level whenever appropriate.

Line 310:

ELISA of inflammatory cytokines and growth factors…..

Specify, lung tissue concentration, plasma concentration?

Line 323

….from the background level of 252.8 � 25.9 to 77.3 � 7.3 on…..

What is the measurement unit? This refers to the entire paragraph!

Line 328

…. level of 87 � 5 to 48 � 5 per 100 alveoli on….

87 what? 87 cells per 100 alveoli?

Line 354

….VILI (0.19 � 0.05; p = 0.009, Fig 5) a…

Unit?

Line 379

….. activated by an underlying bacterial infection.

Rephrase. Bacterial infection is just one of many activators.

Line 391

The expression of Gr1 (Ly6C) on monocytes is closely linked to their divergent roles in inflammatory tissue.

Avoid nonsensical phrases like: linked ….to divergent roles.

Line 386

…VILI, contrary to inflammatory cytokines.

That is not contrary. VEGF increases while inflammation fades or vice versa. Rephrase.

Line 386

VEGF expression in lung tissue is associated with the proliferation of alveolar epithelial cells.

That is an assumption or quote of former works.

First state the main findings. Then, refer to existing knowledge. Then, present a conclusion.

7. PLOS authors have the option to publish the peer review history of their article (what does this mean?). If published, this will include your full peer review and any attached files.

Reviewer #1: No

Reviewer #2: No

---

## [Author Response · Author response to Decision Letter 1]

19 Feb 2021

Chung-Sheng Shi

Chang Gung University

Graduate Institute of Clinical Medicine Sciences

No. 259, Wenhua 1st Rd

Guishan Dist., Taoyuan City 33302, Taiwan

csshi@mail.cgu.edu.tw

Dr. Heinz Fehrenbach

Academic Editor

PLOS ONE

February 17, 2021

Subject: Revision and resubmission of manuscript PONE-D-20-27843 

Dear Dr. Heinz Fehrenbach,

Thank you for your letter and the opportunity to revise our paper in PLOS ONE again. In response to the reviewers’ comments, we have detailed point-by-point responses to all comments along with our revised manuscript.

I have included the comments immediately after this letter and responded to them individually, indicating exactly how we addressed each concern or problem and describing the changes we have made. Besides, we have had our manuscript professionally edited by Editage again. All four authors have approved the revisions, and I have again been chosen as the corresponding author. The changes are marked in red in the paper as you requested, and the marked and unmarked versions of the revised manuscript have been uploaded to the online submission system.

We hope the revised manuscript will better suit PLOS ONE but are happy to consider further revisions, and we thank you for your continued interest in our research.

Sincerely,

Chung-Sheng Shi

Graduate Institute of Clinical Medicine Sciences, College of Medicine, Chang Gung University,

 

Author Responses to Reviewers and Manuscript Changes

Response to Reviewers

Thank you for your precious comments and advice. Those comments are valuable and helpful for revising and improving our paper and the critical guiding significance to our research. We have studied words carefully and have made corrections, which we hope meet with approval. Revised portions are marked in red on the paper. The major corrections in the paper and the responses to the reviewer’s comments are as following:

Reviewer #1: The resubmitted manuscript entitled „Roles of lung-recruited monocytes and pulmonary vascular endothelial growth factor (VEGF) in resolving ventilator-induced lung injury (VILI)” by Chung-Sheng Shi et al. describes the role of monocytes and the release of VEGF in a model of VILI. The authors described

All points of the reviewer are properly edited the statistics and described the different statistical models in supplement. All numbers of the experiments were mentioned and statistical variances shown as mean � SD. The text body was well corrected and most of the typos removed. There are some miner points I would like to address.

Response:

Thank you for your summary. We appreciate your efforts in reviewing our manuscript. We have revised the manuscript accordingly. Our point-by-point responses are detailed below:

Minor points:

1. In figure 4B, day 1 the authors should check if the magnification is really the same as in the other histological images. It seems to be a higher magnification. Please clarify.

Response:

We have confirmed that the magnification of the day 1 image in figure 4B is the same as that of other images. The alveoli in this image may be damaged and expanded due to VILI, so it is why the alveoli' size in this image is different from others. However, to respect the inflammatory cells, the size is the same.

2. The correlation between the pulmonary VEGF and alveolar epithelial barrier one side and the role of monocytes on the other side are two possible explanations for the changes of VEGF during VILI. However, the authors work out the differences more detailed in the discussion.

Response:

Thank you for your comments. We are very grateful for your affirmation.

3. In the discussion section (line 378-381) the authors compared the bacterial infection in ventilated patients with their two-hit model of LPS and ventilation. The author should clarify that there is a substantial difference between bacterial infection and LPS treatment in terms of cytokine profiles

Response:

Thank you for your precious comments and advice. We have rewritten the first paragraph (lines 407-416) and discussed the substantial difference between bacterial infection and LPS treatment in the second paragraph (lines 417-428). In the paragraph, we have discussed the substantial difference of the time-course expression of inflammatory cytokines between bacterial infection and LPS treatment and cited new references (line 423).

Reviewer #2: General comments.

Please, clearly decribe how many animals were used in each Group. How many animals entered a Group, how many survived VILI, how many were sacrificed early, etc. Discuss, speculate how this may have impacted on the results? Did animals with an excessive Inflammation die early? Did you check?

Consider a figure, flow chart.

Response:

Thank you for your precious comments and advice. We have made a flow chart in supporting information S1 Fig, shown below, to illustrate the numbers of animals used in the different experimental groups and the numbers of animals that died during mechanical ventilation and recovery. For some of the experimental mice died before being sacrificed. The cause of death may have been excessive inflammation and lung injury. This suggests that the concentrations of inflammatory cytokines in the injured lung tissue may have been underestimated. We have stated the situation as a limitation of the study in the discussion section (lines 541-545).

S1 Fig. Flow chart depicting animal experiment design.

Please, clearly decribe the ventilation used to induce VILI.

Response:

Thank you for your advice. We have rewritten the section of materials and methods and described the detail of the ventilation set for inducing VILI (lines 102-111).

Please, clearly describe the anima handling after the experiment. Did the mice receive pain medication, etc.

Response: 

Thank you for your careful review. We have described the detail of the animal handling after the experiment in the section of materials and methods ( lines 111-115). The experimental mice did not receive pain medication after the experiment for no tracheostomy creation for intubation and no other wound to be produced by the experimental procedures.

The statistics used may not be appropriate. You used parametric tests despite the low n. The use of non-parametric test would be more cautious. Use non-parametric tests or explain. Did you check for normal distribution in all groups?

Response:

Thank you for your precious comments and advice. We have rechecked the normality of the data by using Shapiro-Wilk test and used parametric or non-parametric tests as appropriate. For comparisons between two groups, data were analyzed by unpaired Mann–Whitney U test. For comparisons between multiple groups, parametric (one-way ANOVA with Bonferroni’s or Dunnett’s post hoc test) or non-parametric test (the Kruskal-Wallis test with Bonferroni correction) was used. Besides, we used Spearmen’s correlation to evaluate the relationship between the IHC intensity of VEGF and the number of Ki67-positive cells in the alveolar epithelium of lung sections (lines 231-236). Detailed illustrations of the statistical analysis are provided in the supporting information S1 Text.

Avoid nonsensical phrases: The results reveal, etc.

Response:

Thank you for your advice. We have rewritten the phrases (lines 275 and 379).

Always state the measurement unit.

Response:

Thank you for your careful review and advice. We have rechecked and made sure that the measurement units have been stated. Besides, for the statements of the IHC staining intensity and mRNA expression, we have presented with relative intensities and fold, respectively.

Line 79

Animals and two-hit model of VILI

The description of the animal model is insufficient.

Clearly describe each hit of the two hit model, e.g.:

The first hit consisted of ……..

The second hit ……..

Then, give a detailed report of the experimental procedure in the correct chronological order.

i. anesthesia

ii. LPS-injection

iii. Intubation

iv. Ventilation

v. Extubation

Specify, the ventilation used to induce VILI besides tidal volume and respiratory rate further: inspiration time, I:E, resulting peak pressure and duration of ventilation!

Whan did you stop ventilation. How and when as a mouse extubated? Did you extubate right from VILI ventilation? What kind of ventilation did you used to wean anesthesia.

Response:

Thank you for your precious comments and advice. We have rewritten and detailed the two-hit model's description in the section of materials and methods (lines 95-111). 

Line 92:

….water, and the center was maintained in low noises and vibrations.

Sounds awkward. Rephrase

Response:

Thank you for your comment. We have rewritten the paragraph (lines 89-93).

Line 112:

The number of animals used per group was as follows: normal control (n = 6), day 0 (n = 6 ), day 1 (n = 6), day 3 (n = 6), day 7 (n = 12, 6 of them for the study of depletion of pulmonary monocytes), and day 14 (n = 12, 6 for the study of depletion of pulmonary monocytes).

This sounds to good. Not one animal died? How many animals did you loose during induction of VILI. How many animals entered each group and how many remained over the following days? What was the selection process to euthanize an animal prematurely?

It needs to be discussed in the discussion section in how far that may have altered the results.

Response:

Thank you for your precious comments and advice. We have detailed in a flow chart, shown in S1 Fig, to illustrate the numbers of animals used in the different experimental groups and the numbers of animals that died during mechanical ventilation and recovery. For some of the experimental mice died before being sacrificed. The cause of death may have been excessive inflammation and lung injury. This suggests that the concentrations of inflammatory cytokines in the injured lung tissue may have been underestimated. We have stated the situation as a limitation of the study in the discussion section (lines 541-545).

Line 117:

How were the lungs inflated – manually? With a defined pressure/volume?

Response:

Thank you for your careful review. We have rewritten the section of materials and methods. For histopathology and immunohistochemistry analysis, mouse lungs were manually inflated and fixed by injecting 10% buffered formalin and embedded in paraffin (line 123).

Line 215:

How were the mice sedated for this injection. Did you need to ventilate them?

Response:

Thank you for your careful review. We have rewritten and detailed the description of mouse anesthesia in the experiment of “Depletion of pulmonary monocytes” ( lines 225-226 and 228). The experimental mice received injection with Clophosome-A in the tail vein after anesthesia with intraperitoneal Zoletil 50. During the experiment, the mice remained with spontaneous breath and did not receive mechanical ventilation.

Line 222:

You used parametric tests despite the small number of animals. Did you check for normal distribution? Non-parametric tests and Spearman correlation may be the more conservative strategy.

Response:

Thank you for your precious comments and advice. We have rechecked the normality of the data and used Spearman correlation to assess association (line 236).

Line 245:

C57BL/6 mice received i.v. with 20 ng of LPS and high-stretch mechanical ventilation that produced two-hit VILI.

Sentence does not make sense. Rephrase.

Response:

Thank you for your comment. We have rewritten the sentence (lines 258-259).

Line 253:

….; n = 3-5, ….

What does that n mean? It does not match the group size given in the method section?

This refers to all other n = ..... in the figure descriptions.

Response:

Thank you for your careful review. We have correct the typing error in the section of materials and methods ( line 143) and removed the wrong description in the section of results (line 254). Detailed illustrations of the statistical analysis are provided in the supporting information S1 Text.

Line 262:

Avoid nonsencical phrases like: The results revealed ……

Response:

Thank you for your advice. We have rewritten the sentence (line 275).

Line 281:

..SD; n = 4-5,

What is this n?

Response:

Thank you for your careful review. For flow cytometric analysis, the number of animals used per group was as follows: normal control (n = 4), day 0 (n = 4), day 1 (n = 4), day 3 (n = 4), day 7 (n = 5), and day 14 (n = 5), which has been stated in the section of materials and methods. We have correct the descriptions in the legend of Fig 2 (lines 295 and 299).

General comment:

What is a background level of something in mice not held under experimental conditions?

If you mean the first measured level before the intervention you may want to use the term baseline. Do you mean the level in the control animals at that specific time point?

Consider using concentration instead of level whenever appropriate.

Response:

Thank you for your precious comments and advice. We have corrected the descriptions to use the term “baseline” instead of “background level” and “concentration” instead of “level” accordingly.

Line 310:

ELISA of inflammatory cytokines and growth factors…..

Specify, lung tissue concentration, plasma concentration?

Response:

Thank you for your careful review. We have detailed the description to clarify that the ELISA samples were obtained from the lung tissue of mice (line 328). 

Line 323

….from the background level of 252.8 � 25.9 to 77.3 � 7.3 on…..

What is the measurement unit? This refers to the entire paragraph!

Response:

Thank you for your careful review and comments. We have correct the measurement unit as the relative staining intensity (lines 341, 345, 354, and 368).

Line 328

…. level of 87 � 5 to 48 � 5 per 100 alveoli on….

87 what? 87 cells per 100 alveoli?

Response:

Thank you for your careful review. We have correct the description of the unit (lines 348 and 352). 

Line 354

….VILI (0.19 � 0.05; p = 0.009, Fig 5) a…

Unit?

Response:

Thank you for your careful review. We have correct the description of the unit (lines 380, 381, and 383).

Line 379

….. activated by an underlying bacterial infection.

Rephrase. Bacterial infection is just one of many activators.

Response:

Thank you for your precious comments and advice. We have rewritten the sentence (line 417).

Line 391

The expression of Gr1 (Ly6C) on monocytes is closely linked to their divergent roles in inflammatory tissue.

Avoid nonsensical phrases like: linked ….to divergent roles.

Response:

Thank you for your careful review and comments. We have rewritten the sentence (lines 430-431).

Line 386

…VILI, contrary to inflammatory cytokines.

That is not contrary. VEGF increases while inflammation fades or vice versa. Rephrase.

Response:

Thank you for your precious comment and advice. We have rewritten the paragraph (lines 411-412). 

Line 386

VEGF expression in lung tissue is associated with the proliferation of alveolar epithelial cells.

That is an assumption or quote of former works.

First state the main findings. Then, refer to existing knowledge. Then, present a conclusion.

Response:

Thank you for your precious comments and advice. We have rewritten the description that the main finding that the VEGF expression in lung tissue is positively related to the expression of Ki67, a proliferation marker, in alveolar epithelial cells (lines 412-414). We have also discussed the finding and possible VEGF roles in post-injury lung referred from other studies (lines 485-512). Finally, we conclude that pulmonary VEGF may play a crucial role in post-VILI lung repair (line 568).

---

## [Decision Letter · Decision Letter 2]

9 Mar 2021

Roles of lung-recruited monocytes and pulmonary vascular endothelial growth factor (VEGF) in resolving ventilator-induced lung injury (VILI)

PONE-D-20-27843R2

Dear Dr. Shi,

We’re pleased to inform you that your manuscript has been judged scientifically suitable for publication and will be formally accepted for publication once it meets all outstanding technical requirements.

Kind regards,

Heinz Fehrenbach

Academic Editor

PLOS ONE

Additional Editor Comments (optional):

Reviewers' comments:

Reviewer's Responses to Questions

**Comments to the Author**

1. If the authors have adequately addressed your comments raised in a previous round of review and you feel that this manuscript is now acceptable for publication, you may indicate that here to bypass the “Comments to the Author” section, enter your conflict of interest statement in the “Confidential to Editor” section, and submit your "Accept" recommendation.

Reviewer #1: All comments have been addressed

Reviewer #2: All comments have been addressed

2. Is the manuscript technically sound, and do the data support the conclusions?

Reviewer #1: Yes

Reviewer #2: Yes

3. Has the statistical analysis been performed appropriately and rigorously? 

Reviewer #1: Yes

Reviewer #2: Yes

4. Have the authors made all data underlying the findings in their manuscript fully available?

Reviewer #1: Yes

Reviewer #2: Yes

5. Is the manuscript presented in an intelligible fashion and written in standard English?

Reviewer #1: Yes

Reviewer #2: Yes

6. Review Comments to the Author

Reviewer #1: The resubmitted manuscript entitled „Roles of lung-recruited monocytes and pulmonary vascular endothelial growth factor (VEGF) in resolving ventilator-induced lung injury (VILI)” by Chung-Sheng Shi et al. describes the role of monocytes and the release of VEGF in a model of VILI. The authors described

All points of the reviewer are properly addressed. The statistics methods and conclusions are valid.

The interpretation of the LPS response in the VILLI-model was distinguished from an a bacterial infection in the discussion section.

In addition the experimental model is described in more detail, which approved the manuscript. Even, the flowchart for the experimental design is very helpful.

Reviewer #2: Thank you for your detailed and lengthy response to all of my questions.

It is my humble opinion, that the article has markedly improved.

7. PLOS authors have the option to publish the peer review history of their article (what does this mean?). If published, this will include your full peer review and any attached files.

Reviewer #1: No

Reviewer #2: No

---

## [Editor Report · Acceptance letter]

11 Mar 2021

PONE-D-20-27843R2 

Roles of lung-recruited monocytes and pulmonary vascular endothelial growth factor (VEGF) in resolving ventilator-induced lung injury (VILI) 

Dear Dr. Shi:

I'm pleased to inform you that your manuscript has been deemed suitable for publication in PLOS ONE. Congratulations! Your manuscript is now with our production department. 

Kind regards, 

on behalf of

Prof. Dr. Heinz Fehrenbach 

Academic Editor

PLOS ONE